# Brain Targeting by Intranasal Drug Delivery: Effect of Different Formulations of the Biflavone “Cupressuflavone” from *Juniperus sabina* L. on the Motor Activity of Rats

**DOI:** 10.3390/molecules28031354

**Published:** 2023-01-31

**Authors:** El-Sayed Khafagy, Gamal A. Soliman, Ahmad Abdul-Wahhab Shahba, Mohammed F. Aldawsari, Khalid M. Alharthy, Maged S. Abdel-Kader, Hala H. Zaatout

**Affiliations:** 1Department of Pharmaceutics, College of Pharmacy, Prince Sattam bin Abdulaziz University, Al-kharj 11942, Saudi Arabia; 2Department of Pharmaceutics and Industrial Pharmacy, Faculty of Pharmacy, Suez Canal University, Ismailia 41522, Egypt; 3Department of Pharmacology, College of Pharmacy, Prince Sattam Bin Abdulaziz University, P.O. Box 173, Al-Kharj 11942, Saudi Arabia; 4Department of Pharmacology, College of Veterinary Medicine, Cairo University, Giza 12211, Egypt; 5Kayyali Chair for Pharmaceutical Industries, Department of Pharmaceutics, College of Pharmacy, King Saud University, P.O. Box 2457, Riyadh 11451, Saudi Arabia; 6Department of Pharmacognosy, College of Pharmacy, Prince Sattam Bin Abdulaziz University, P.O. Box 173, Al-Kharj 11942, Saudi Arabia; 7Department of Pharmacognosy, Faculty of Pharmacy, Alexandria University, Alexandria 21215, Egypt

**Keywords:** *Juniperus sabina*, cupressuflavone, intranasal, pharmacokinetic, motor coordination

## Abstract

The polar fractions of the *Juniperus* species are rich in bioflavonoid contents. Phytochemical study of the polar fraction of *Juniperus sabina* aerial parts resulted in the isolation of cupressuflavone (CPF) as the major component in addition to another two bioflavonoids, amentoflavone and robustaflavone. Biflavonoids have various biological activities, such as antioxidant, anti-inflammatory, antibacterial, antiviral, hypoglycemic, neuroprotective, and antipsychotic effects. Previous studies have shown that the metabolism and elimination of biflavonoids in rats are fast, and their oral bioavailability is very low. One of the methods to improve the bioavailability of drugs is to alter the route of administration. Recently, nose-to-brain drug delivery has emerged as a reliable method to bypass the blood–brain barrier and treat neurological disorders. To find the most effective CPF formulation for reaching the brain, three different CPF formulations (A, B and C) were prepared as self-emulsifying drug delivery systems (SEDDS). The formulations were administered via the intranasal (IN) route and their effect on the spontaneous motor activity in addition to motor coordination and balance of rats was observed using the activity cage and rotarod, respectively. Moreover, pharmacokinetic investigation was used to determine the blood concentrations of the best formulation after 12 h. of the IN dose. The results showed that formulations B and C, but not A, decreased the locomotor activity and balance of rats. Formula C at IN dose of 5 mg/kg expressed the strongest effect on the tested animals.

## 1. Introduction

The frequency of neurological illnesses has been rising over the past few years. A billion people worldwide may suffer from neurological illnesses, according to a 2020 World Health Organization (WHO) assessment [1]. Based on this fact, neurological illnesses are regarded as one of the leading causes of disability and fatalities globally [1,2]. Despite the fact that the majority of novel entities never make it to clinical trials, scientific efforts are continuous to find and create new and effective neuropharmaceuticals.

In fact, successful medications should be able to pass via absorptive membranes, escape the hepatic first-pass effect, and then cross the blood–brain barrier (BBB) unaltered to reach the brain [3,4]. The drug molecule must be lipophilic, with low molecular weight (400 Da), and nonionizable at physiological pH in the absence of the active efflux transporter mechanism to pass through the BBB [3,5]. The lack of such characters reduces the drug bioavailability by limiting the solubility and absorption rate leading to diminished effectiveness [6,7]. Therefore, substantial dosages must be administered to reach minimal effective blood concentrations [8].

Alternative formulations have been created to address the issues, with lipidic nanosystems garnering greater attention recently as nanoemulsions, solid lipid nanoparticles, nano-lipid carriers, and self-emulsifying drug delivery systems (SEDDS) [6]. These systems’ major objective is to maintain lipophilic chemicals in solution following contact with aqueous environments, such as those found in the nasal mucosa or gastrointestinal (GI) tract [6]. Lipophilic BCS class II and IV medicines can be included into self-emulsifying drug delivery systems (SEDDS), a form of lipidic nanosystem that is well known for this property [6,7]. The oral route has been the focus of the majority of SEDDS research up until this point [9].

Other delivery methods, however, may be of significant interest for SEDDS, especially if they enhance brain targeting of CNS-active medications. In this situation, it might be advantageous to investigate the intranasal (IN) administration of medications included in lipidic SEDDS. The nasal cavity is the only anatomical region that directly connects the central nervous system (CNS) with the outside world, which is the fundamental justification. Because medicines can partially cross the BBB and enter the brain directly, this mode of delivery becomes particularly appealing for treating neurological illnesses [10,11]. Drugs delivered through the nasal cavity can also pass through the bloodstream to the brain. As a result, there is no GI passage, no hepatic first-pass effect, and granting systemic drug absorption [10,12,13]. Combining the IN route as a potential conduit for nose-to-brain administration with the development of neurotherapeutics in SEDDS form can lead to a considerable rise in brain bioavailability, and an advancement in patient therapeutic management may be anticipated. In addition, numerous long-term side effects of illnesses and non-adherence to medication may be prevented in this way [8].

Genus *Juniperus* is a member of the cypress family (Cupressaceae). The genus comprises more than 65 species widely distributed in the Northern Hemisphere [14]. Members of the genus are evergreen shrubs or trees with needle- or scale-like leaves [15]. Previous phytochemical studies of the *Juniperus* species resulted in the isolation of diterpenes, sesquiterpenoids, lignans, phenylpropanoid, flavonoids, and coumarins [16,17,18]. Several biflavonoids including amentoflavone, cupressuflavone, robustaflavone, hinokiflavone, and mono-O-methylhinokiflavone were isolated from the polar fractions of the leaf extract of *J. phoenicea* [19,20,21]. 

Amentoflavone and related bioflavonoids were reported as potent psychoactive drug leads. Their effect is due to the interactions with CNS receptors and biogenic amine transporters. Biflavonoids such as amentoflavone and dihydroamentoflavone showed the strongest binding activity to rBZP, GABAA receptor, and NET [22]. Biflavonoids including cupressuflavone demonstrated neuroprotective effects via the inhibition of β-secretase and cyclin-dependent kinases (CDKs) [23]. Naturally occurring biflavonoids were also effective against oxidative stress-induced and amyloid β-peptide-induced cell death in neuronal cells [24]. Amentoflavone and ginkgetin from the leaves of *Chamaecyparis obtusa* significantly protected HT22 cells against glutamate-induced oxidative stress [25].

Accordingly, the aim of the current study is to formulate cupressuflavone in a self-nanoemulsifying drug delivery system (CPF-SNEDDS) for IN administration to achieve combined enhancement of drug solubility and cross the BBB to improve CNS bioavailability. The drug was initially incorporated in an optimized SNEDDS formulation, which was then evaluated in vivo to estimate the pharmacological and pharmacokinetic parameters. 

## 2. Results and Discussion

### 2.1. Characterization of the Isolated Compounds

HRESIMS of the three isolated compounds (Appendix A) showed (M+H)^+^ at *m*/*z* 539.0972, 539.0981, and 539.0973 for **1**, **2,** and **3**, respectively, all represented the molecular formula C_30_H_18_O_11_. The UV data of **1–3** was diagnostic for the 5, 7, 4′-trihydroxy flavone skeleton [26]. The ^1^H- and ^13^C-NMR (Appendix A) of **1** showed a set of signals for one apigenin skeleton lacking the H-8 resonance [27]. The MS and NMR data of **1** were diagnostic for symmetric biflavone. Literature data and direct comparison with a previously isolated sample enabled the identification of **1** as cupressuflavone (Figure 1) [21,28].

Compounds **2** and **3** showed signals for two flavonoidal skeletons indicating asymmetric dimeric structures. The ^1^H-NMR of **2** and **3** (Appendix A) showed ABX coupling systems at δ_H_ 8.32 (d, J = 1.8 Hz), 6.86 (d, J = 8.6 Hz), and 7.87 (dd, J = 1.8, 8.6 Hz) and 8.18 (bs), 6.79 (d, J = 8.3 Hz), 7.66 (bd, J = 8.3 Hz) assigned for disubstituted ring-B in part II of **2** and **3**, respectively. The values of the carbon chemical shifts of C-3′ and C-4′ at δ_C_ 124.3, 160.8 and 119.9, 158.5 ppm in **2** and **3**, respectively, indicated that C-3′ is not oxygenated and it is the point of connection with part I of the two compounds. Both **2** and **3** lack either H-6 or H-8 signal of part I in the ^1^H-NMR. The data of **2** were identical with those reported for amentoflavone (Figure 1) [28,29] with connection to part II via C-8. Compound **3** was confirmed as the skeleton where connection to part II took place via C-6. The data of **3** enable its identification as robustaflavone (Figure 1) [29,30].

### 2.2. Excipients Selection and Formulation Optimization

The preliminary precipitation-based solubility study showed that CPF is highly soluble in DMSO while it showed poor solubility in several other excipients with diverse chemical compositions ranging from medium to long chain, monoglycerides/triglycerides/free fatty acids along with several co-solvents and surfactants (Table 1). These findings reflect the challenging physicochemical properties of CPF that limit its efficient formulation. According to these findings, DMSO was selected as an essential excipient within all the selected formulations. On the other hand, Pluronic F127 and Cremophor El were selected to formulate CPF due to their emulsification capabilities along with their potential role in nose-to-brain delivery, as previously demonstrated [31,32,33].

The self-emulsifying nature of the various formulations upon aqueous dispersion showed different degrees of transparency ranging from a clear to whitish milky suspension. The formulations were optimized based on their self-emulsification efficiencies. The compositions of the optimized formulations are presented in Table 2. The most interesting formulation system (formula C) was able to produce nanoemulsions (SNEDDS) (Table 2). Components were developed using water-soluble surfactants and co-solvents.

### 2.3. Formulation Characterization

Both formulations B and C showed (<200 nm) nano-scale particle size upon aqueous dilution. However, formulation C showed significantly lower droplet size and polydispersity index (PDI) compared to formulation B (*p* < 0.05) (Figure 2A,B). Formulation C showed an average of 37 nm droplet size and 0.2 PDI which suggests the ultimate efficiency of the self-nanoemulsification process and could be attributed to the presence of the highly hydrophilic surfactant (Cr-El) in the formulation. Several previous studies confirmed the crucial effect of hydrophilic surfactants (high HLB) on the droplet size and self-emulsification efficiency of SNEDDSs [34,35]. 

On the other hand, both formulations B and C showed low zeta potential values (−6 and −4 mV, respectively) (Figure 2C). Zeta potential is an important parameter that could be linked with nanoemulsion stability. Colloids with low zeta potential absolute values (negative or positive) are prone to the risk of particle agglomeration and physical instability upon storage. Accordingly, formulation C was designed to be prepared and stored in an anhydrous form while it was mixed with water only prior to intranasal administration to avoid undesirable changes in formulation stability upon storage.

### 2.4. Effect on Spontaneous Motor Activity

Figure 3 shows the effect of Chlorpromazine (CPZ), and A, B, and C formulae on the spontaneous motor activity as vertical and horizontal beam breaks by the rats. Horizontal activity measures exploratory activities at the floor level of the activity cage and small body movement activity such as grooming. On the other hand, vertical activity measures rearing and elevated sniffing movements. Within 10 min observation, the horizontal and vertical activities of rats treated with CPZ, formula B, and C were significantly decreased relative to the NC rats. Further, there were no significant differences in the horizontal and vertical activities between the formula A-treated group and NC rats. The effects of formula C on the horizontal and vertical activities of rats were 2.43- and 2.27-fold, respectively, lower than those observed in the NC group.

### 2.5. Effect on Motor Coordination and Balance

The effects of CPZ, A, B, and C formulae on the motor coordination and balance of rats using the accelerating rotarod test are displayed in Figure 4. Based on this test, we observed no significant difference in latency to fall between the A-treated rats (259.2 ± 9.57 s) and NC group (286.5 ± 9.84 s). However, the motor performance declined in rats exposed to CPZ and formula B and C in comparison with the NC group. Rats treated with C formula fell off the rotarod faster (115.8 ± 5.22 s) than those treated with formula A (259.2 ± 9.57 s) and B (194.6 ± 6.36 s). Moreover, formula C showed a 2.47-fold decrease in motor coordination and balance compared to the NC group.

### 2.6. Systemic Absorption of CPF after Intranasal Administration

Biflavonoids and flavonoids may be transported from the systemic blood to the brain through the BBB with possible CNS activation [36,37]. Rats that had undergone Hirai’s surgical procedure were given the drug solution and its optimized formula via the intranasal route to assess the systemic absorption of CPF. Figure 5 shows the time profiles of systemic absorption of CPF after the intranasal administration of drug and its optimized formula C. CPF was slightly absorbed after the nasal administration in the form of solution. The intranasal administration of CPF-optimized formula C significantly increased the nasal absorption and the plasma concentration that was elevated at 15 min post intranasal administration.

Table 3 provides a statistical summary of the pharmacokinetic parameters for the CPF-optimized formula and CPF solution. The CPF level reached its maximal concentration in the plasma after 0.5 h following the intranasal administration of both the drug solution and the CPF-optimized formula, according to the pharmacokinetics results (formula C). This Tmax might be attributed to the fast absorption through the intranasal route. The plasma Cmax of the intranasal CPF formula was 422.5 ± 54.71 ng/mL, and the intranasal solution was 29.16 ± 3.56 ng/mL. The AUC_0–12_ was 263.02 ± 62.29 ng.h/mL for the CPF solution and 29.274.87 ± 4.855.87 ng.h/mL for the optimized CPF formula. Statistics indicated that these pharmacokinetic parameter changes were significant. This implied that the systemic absorption of CPF following intranasal delivery was optimal with CPF formula C.

### 2.7. In Vivo Brain Distribution of CPF after an Intranasal Administration

Considering that CPF is known to cross the BBB, a rise in plasma levels of the substance could lead to its distribution into the brain. We showed that giving the improved formula via the intranasal route enhanced the body’s absorption of CPF (Figure 4 and Table 3), encouraging further estimation of the CPF amount in the brain. 

The CPF formula was validated via the brain concentration of CPF at 15 min after intranasal injection (Figure 6). The findings imply that intranasal delivery of the CPF formula tended to improve the distribution of CPF into the brain. The optimized formula preferentially enhanced the brain concentration of CPF. These results may be due to making direct transportation of CPF from the nasal cavity to the brain easier. The findings thus revealed that intranasal administration of the formula could enhance the transfer of CPF to the brain. In accordance, previous studies revealed that the addition of a hydrophilic emulsifying agent such as Cremophor RH40 increased fluorescein isothiocyanate brain uptake [36]. Pluronic block copolymers such as Pluronic^®^ P85 showed the ability to increase the blood–brain barrier permeability of several drugs [32]. In addition, Pluronic^®^ F127 was reported to enhance the mucus penetration of nanoparticles for nose-to-brain delivery [33].

## 3. Materials and Methods

### 3.1. General

Melting points were measured using open capillary tubes *Thermosystem FP800 Mettler FP80* central processor supplied with *FP81 MBC* cell apparatus and were uncorrected. Ultraviolet absorptions were obtained using a Unicum Heyios a UV–Visible spectrophotometer. ^1^H-, ^13^C-NMR, and 2D-NMR experiments were collected using UltraShield Plus 500 MHz (Bruker, Billerica, MA, USA) (NMR Unite at the College of Pharmacy, Prince Sattam Bin Abdulaziz University) spectrometer operating at 500 MHz for protons and 125 MHz for carbon atoms, respectively. Chemical shift values are reported in δ (ppm) relative to the residual solvent peak, and the coupling constants (*J*) are reported in Hertz (Hz). HRMS were determined by direct injection using Thermo Scientific UPLC RS Ultimate 3000—Q Exactive hybrid quadrupole-Orbitrap mass spectrometer combining high performance quadrupole precursor selection with high resolution, accurate-mass (HR/AM) Orbitrap™ detection. Direct infusion of isocratic elution acetonitrile/methanol (70:30) with 0.1% formic acid was used to flush the samples. Runtime was 1 min using nitrogen as auxiliary gas with flow rate 5 μL/min. Scan range from 160–1500 *m*/*z* was used. Resolving power was adjusted to 70,000 @ *m*/*z* 200. Detection was in both positive and negative modes separately. Calibration was conducted using Thermo Scientific Pierce™ LTQ Velos ESI Positive Ion Calibration Solution including Caffeine, Met-Arg-Phe-Ala (MRFA), Ultramark 1621, n-Butyl-amine components, and Pierce™ LTQ Velos ESI. Negative Ion Calibration Solution includes sodium dodecyl sulphate (SDS), sodium taurocholate, Ultramark 1621 components. Capillary temperature set at 320 °C and capillary voltage at 4.2 Kv. Sephadex LH-20 (Amersham Biosciences, Uppsala, Sweden), silica gel 60/230–400 mesh (EM Science, Swedesboro, NJ, USA), RP C-18 silica gel 40–63/230–400 mesh (Fluka) was used for column chromatography. The thin layer chromatography (TLC) analysis was performed on Kiesel gel 60 F254 and RP-18 F254 (Merck, Rahway, NJ, USA) plates. A UV lamp (entela Model UVGL-25) operated at 254 nm was used for detecting spots on the TLC plates.

### 3.2. Plant Materials

Aerial parts of *Juniperus sabina* L. (Cupressaceae) were described earlier [38].

### 3.3. Extraction, Fractionation and Purification

The dried ground aerial parts (1000 g) were extracted till exhaustion by percolation at room temperature with 95% ethanol (10 L). The resulted extract was concentrated in vacuo to leave dark green viscous residue with aromatic odor. Approximately 200 g of the total extract were dissolved in 1200 mL of 20% aqueous ethanol and fractionated with petroleum ether (500 mL × 3) to yield 45.86 g petroleum ether soluble fraction. The aqueous ethanol fraction was diluted with water to increase water content to 40% and the resulted solution was fractionated with chloroform (500 mL × 3) to yield 44.93 g of chloroform soluble fraction, while the remaining aqueous ethanol soluble layer was lyophilized to produce 98.54 g. 

The lyophilized aqueous ethanol fraction was reconstituted in water and filtered to yield 37.54 g of water insoluble fraction. Part of the water insoluble fraction (20 g) was purified over Sephadex LH-20 column (300 g, 5 cm id) eluting with methanol. Fractions of 20 mL each were collected, screened with TLC and similar fractions were pooled. Fractions 14–28 provided 435 mg of **1**. Fractions 36–42 yielded 174 mg of **2**, while fractions 50- 52 afforded 8 mg of **3**. 

### 3.4. Characterization of the Isolated Compounds

**Cupressuflavone** (**1)**: Yellow powder; mp 392 °C Decomp.; UV λ_max_ (MeOH) 329, 276, and 226 nm; ^1^H- and ^13^C-NMR (CD_3_OD, δ): Appendix A. HRESIMS: *m*/*z* 539.0972 (*Cal*. 539.0978 for C_30_H_18_O_11_) (100, [M+H]^+^).

**Amentoflavone** (**2)**: Yellow powder; mp 260 °C; UV (λ._max_, MeOH): 335, 271, 224 nm; ^1^H- and ^13^C-NMR (CD_3_OD, δ): Appendix A. HRESIMS: *m*/*z* 539.0981 (*Cal*. 539.0978 for C_30_H_18_O_11_) (100, [M+H]^+^).

**Robustaflavone** (**3)**: Yellow powder; mp 351 °C; UV (λ._max_, MeOH): 333, 268, 225 nm; ^1^H- and ^13^C-NMR (CD_3_OD, δ): Appendix A. HRESIMS: *m*/*z* 539.0973 (*Cal*. 539.0978 for C_30_H_18_O_11_) (100, [M+H]^+^).

### 3.5. Estimation of Apparent Drug Solubility

Several excipients including oils, co-solvents, co-surfactants, and surfactants were preliminarily screened for their suitability for CPF formulation based on their estimated apparent drug solubility. The apparent solubility was indirectly estimated by evaluating drug precipitation-tendency in each excipient. Approximately 1–2 *w*/*w*% CPF was added to each excipient, vortexed to achieve maximum solubilization, and then centrifuged to assess drug precipitation tendency. Excipients that were able to maintain the initial drug amount solubilized were subjected to additional drug loadings until precipitation was observed.

### 3.6. Preparation of CPF Formulations for Intranasal Delivery

Three cupressuflavone (CPF) formulations were prepared to deliver the drug intranasally, namely the drug solution, Microwave Irradiation Solid Dispersion (MW-SD), and self-nanoemulsifying drug delivery system (SNEDDS). 

#### 3.6.1. Drug Solution

Pure CPF (5 mg) was dissolved in dimethyl sulfoxide (DMSO) to achieve a final CPF concentration of 0.5% *w*/*w*. The mixture was initially heated at 60 °C, and vortexed to ensure complete homogenization (formulation A). 

#### 3.6.2. Microwave Irradiation Solid Dispersion (MW-SD)

MW-SD was prepared using domestic MW irradiation (Samsung Model ME0113M1). Pure CPF was blended with Pluronic F-127 at 1:19 *w*/*w* ratio (Table 1). A precise amount of the drug and the carrier were gently mixed for approx. 1 min. The MW power was set at power 900 W and the samples were subjected to MW irradiation for 7.5 min. The samples were allowed to cool, solidify, and subsequently pulverized to obtain uniform powder. The prepared MW-SD was then mixed with DMSO, heated at 60 °C, and vortexed to ensure complete homogenization (formulation B). (Table 1) [39].

#### 3.6.3. Solid-Dispersion Loaded Self-Nano Emulsifying Drug Delivery System (SNEDDS)

A predetermined amount of CPF-loaded MW-SD was mixed with DMSO and Kolliphor El, heated at 60 °C, and vortexed to ensure complete homogenization. Prior to assessing the formulation, a precise amount of distilled water was added to the anhydrous formulation as given in Table 1 (formulation C). The mixture was then vortexed for 2 min at ambient temperature to form homogenous mixture [40].

### 3.7. Formulation Characterization

#### Particle Size, Polydispersity Index (PDI), and Zeta Potential

Formulation B and formulation C (anhydrous) were dispersed in Milli-Q water at a ratio of 1:9 *w*/*w*, vortexed for 2 min, and sonicated to ensure uniform formulation dispersion. The diluted formulations were then characterized in terms of the average particle size and polydispersity index (PDI) using a Zetasizer Nano ZS (Malvern Panalytical Ltd., Malvern, UK). Zeta potential was evaluated by laser doppler velocimetry (LDV) mode using the same Nano ZS. Samples were analyzed as triplicates where each replicate was subjected to 3 consecutive measurements (6 runs each).

### 3.8. Experimental Animals

Adult male Sprague Dawley rats with weights ranging from 260 to 280 g were used in this study. Animals were obtained from the lab animal unit at the College of Pharmacy, Prince Sattam bin Abdulaziz University. Animals were kept in ventilated cages; 3–4 rats per cage (IVC Blue Line, Techniplast, Buguggiate VA, Italy). Rats were maintained in controlled laboratory situations (24 ± 0.5 °C and 12 h/12 h dark/light cycle) with feed and water *ad libitum*. The supervision and dealing with rats were in accordance with the international guidelines for use of animals [41]. The study complied with the regulations of the Standing Committee of Bioethics Research at Prince Sattam bin Abdulaziz University which follow the National Regulations on Animal Welfare and Institutional Animal Ethical Committee under the approval number SCBR-048-2022.

### 3.9. Intranasal Delivery of Cupressuflavone Formulations to Rats

Intranasal (IN) delivery of CPF formulations was carried out as described earlier [42]. Briefly, rats were hand-restrained, placed in a supine position, and given the vehicle, chlorpromazine (CPZ), or formulations by IN administration through a micropipette (Pipetman P-20, Gilson Inc., Middleton, WI, USA) in a constant volume of 50 μL [43]. Rats were held in a supine position for 5–10 s after administration to increase the chance for the drug to reach the olfactory region or the nasal cavity with direct access to the brain [44].

### 3.10. Assessment of Spontaneous Motor Activity Using Activity Cage

The activity cage (Model No. 47.420, Ugo Basile S.R.L., Italy) was a Plexiglas box 41 × 41 × 33 (h) cm with 16 infrared (IR) light beams (2.5 cm above the floor level) on the horizontal x- and y-axes, producing a network of perpendicular light beams covering the bottom of the cage for assessment of the horizontal activity. Another set of 16 horizontal IR light beams was elevated 5 cm over the floor plane for measurement of the vertical activity (rearing). The beam interruptions were counted and recorded by an electronic unit connected to the activity cage. The day before measurement of the motor activity, animals were allowed to adapt to the cage apparatus for 60 min. 

On the test day, rats were allowed to further habituate for 30 min, after which IN administrations were delivered. Five groups of 6 rats each were used for the study. The 1st group, which served as a negative control (NC), received the vehicle (saline). The 2nd group received the standard drug, chlorpromazine (CPZ), at a dose of 1 mg/kg and served as a reference group (REF). The test groups (Groups 3, 4, and 5) were given A, B, or C formula, respectively, at 5 mg/kg.

Thirty min after administration, each animal was placed into the activity cage apparatus for 10 min [45,46]. During the experimental period, silent environment was strongly maintained. Spontaneous motor activity was measured by 2 parameters: (a) horizontal activity equivalent to exploratory activities at the floor level of the activity cage and small movement performance (e.g., grooming) and (b) vertical activity measured rearing and high sniffing activities. At the end of the 10 min session, each animal was returned from the activity cage apparatus to its home cage. The inner walls of the activity cage were wiped out with ethyl alcohol (70%) between sessions to block olfactory cues.

### 3.11. Assessment of Motor Coordination and Balance Using Accelerating Rotarod

Motor coordination and balance of the rats were evaluated with the rotarod apparatus (Model No. 7750; Ugo Basile, Italy) following the method reported by Abada et al. [47]. All animals were trained for 3 consecutive days (4 sessions per day). Each session consisted of 4 attempts lasting 300 s with an inter-attempts period of 30 min. During that period, rats were trained to maintain balance against the motion of a rotating rod that increased from 4 to 40 rotations per minute (rpm). Failure of a rat to maintain balance suggests a neurological deficit. Animals that were able to keep their balance on the bar for 180 s were selected and randomly assigned to five groups such as those used in the activity cage test. 

After 30 min of IN administration, each rat was tested 3 times using accelerating speeds of the rotating cylinder of 4 to 40 rpm for 5 min with a 10 min intertrial interval. The device was wiped with ethyl alcohol (70%) and dried prior to each attempt. The mean latency to fall off the rotating cylinder was determined and animals remaining on the rod for more than 300 s were eliminated and their time scored as 300 s.

### 3.12. In Vivo CPF Pharmacokinetic and Biodistribution Study

Rats were confined in a supine position and given an intraperitoneal (i.p.) injection of sodium pentobarbital (50 mg/kg; Nembutal^®^; Abbott Laboratories, Chicago, IL, USA) to induce anesthesia. To keep the anesthetic going, more sodium pentobarbital (12.5 mg/kg) i.p. injections were administered every hour. To close the nasal cavity, surgery was conducted as explained by Hirai et al. [48]. An incision was made in the neck. In order to keep the solutions in the nasal cavity and to maintain respiration, the trachea and esophagus were subsequently cannulated with polyethylene tubing. The nasopalatine ducts were plugged with a medical super glue to stop nasal cavity solutions from draining into the mouth cavity. Rats were given 40 μL (20 μL /nostril) of CPF solution or the chosen formula intranasally using a micropipette (Pipetman P-20, Gilson Inc., Middleton, WI, USA) at a dose of 1 mg/kg body weight (1 mg/mL). Before and 0.25, 0.5, 1, 2, 4, 6, 8, and 12 h after dosage, blood samples (0.25 mL) were obtained from the jugular vein. Heparin was used to heparinize a 1 mL tuberculin syringe by aspirating heparin to cover the syringe wall and then depressing the plunger to the needle hub to release the remaining heparin. The plasma was extracted from the blood by centrifugation at 5400× *g* for 15 min. The plasma sample was stored until analysis at −80 °C in the freezer.

The right jugular vein was used to collect 0.2 mL of blood at predetermined intervals (0.5 h) after injection. The rat’s head was swiftly severed, and the entire brain was carefully isolated, and PBS-cleansed until it was ice cold (pH 7.4). The brain samples were weighed after the water content was removed. The brain samples were homogenized in a homogenizer with two times the volume of ice-cold PBS (pH 7.4) (Heidolph DIAX 900, Chicago, IL, USA). The concentration of CPF was determined using the plasma and supernatant produced after centrifuging the blood sample and brain homogenate at 4 °C and 5400× *g* for 15 min. Until analysis, samples were stored at −80 °C.

### 3.13. LC-MS/MS

Plasma and brain samples were analyzed using UHPLC instrument, equipped with a Quaternary pump, a degasser, and autosampler (Dionex UltiMate 3000, Thermo Fisher Scientific^®^, Waltham, MA, USA). The system is coupled with diode array detector (DAD—3000; Thermo Fisher Scientific^®^). Separation was performed on RP18 HPLC column (150 mm × 4.6 mm i.d., particle size 5 µm, Dionex, Thermo Fisher Scientific^®^), and the column oven was maintained at room temperature. The used mobile phase was composed of ultrapure water (A) and acetonitrile (B), each acidified with 1% acetic acid, under 1.0 mL/min flow rate. An isocratic system composed of 80% A and 20% B run for 20 min was selected for the separation. Concentrations were determined based on isolated CPF detected at 340 nm. From each dilution, a sample of 25 µL was injected using the autosampler. The peak representing CPF was detected at RT = 10.27 ± 0.01.

### 3.14. Pharmacokinetic Parameters

The individual plasma concentration–time curves were used to calculate the maximum plasma drug concentration (C_max_, ng/mL), the time to attain C_max_ (T_max_, h), and the elimination half-life (t_1/2_, h). The area under the curve (AUC) from zero to twelve hours (AUC_0–12_, ng.h/mL) and from zero to infinity (AUC_0–∞_, ng.h/mL) were both determined using the trapezoidal rule approach.

### 3.15. Data and Statistical Analysis

All values were expressed as mean ± standard error of mean (S.E.M.). Statistical analysis was performed using one-way ANOVA with post hoc ’*t*’ test. The statistical analysis was performed using GraphPad Software, San Diego, CA, USA (version 4). Regarding the droplet size, PDI, and zeta potential data, SPSS (version 28) software was utilized to perform the descriptive statistics, identify, and remove the detected outliers. The significance of the data was analyzed by independent *t*-test where *p* values of less than 0.05 were considered as statistically significant.

## 4. Conclusions

Phytochemical study of the polar fraction of *J. sabina* aerial parts extract resulted in the isolation of three known biflavones: cupressuflavone (CPF), amentoflavone, and robustaflavone. The structures were elucidated using spectroscopic methods including UV, 1D, and 2D NMR as well as HRESIMS. The effect of the major component, CPF, on the CNS was demonstrated. For this purpose, three formulae of the compound were developed for intranasal administration as SNEDDSs. The best formulae were developed using a water-soluble surfactant, and co-solvent based on the formulations design of SNEDDSs. In the in vivo evaluation, the rats treated with formula C expressed a decrease in horizontal and vertical movement when tested with the activity cage, and they were unable to maintain their balance when tested with the rotarod apparatus compared to the control group. These findings confirmed the ability of formula C to cause a significant decrease in the motor activity, coordination, and balance of rats after IN administration. Distribution of CPF in the brain and plasma followed the intranasal administration of the developed formula. The pharmacokinetic parameter changes were statistically significant. The data revealed that the optimal CPF formulation involved systemic absorption of CPF after IN administration. The optimized formula boosted the concentration of CPF preferentially, which might indicate the possible direct transport of CPF from the nasal cavity to the brain. 

## Figures and Tables

**Figure 1 molecules-28-01354-f001:**
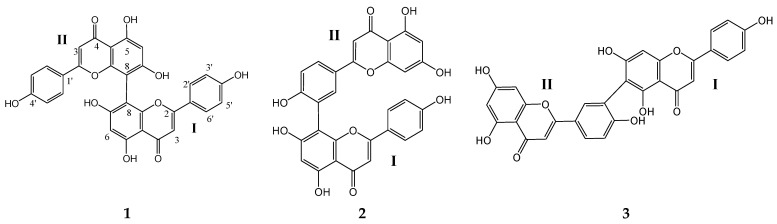
Chemical structure of the isolated biflavones.

**Figure 2 molecules-28-01354-f002:**
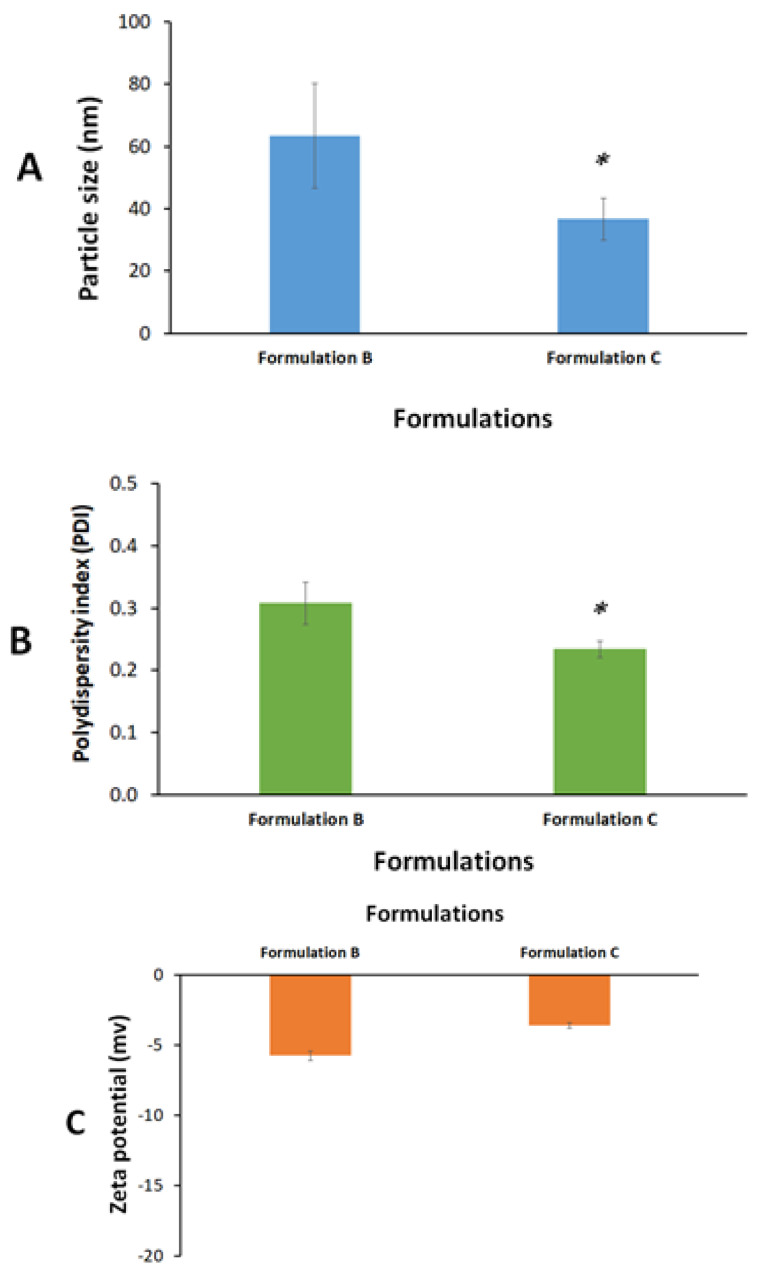
Graphical representations of (**A**) particle size, (**B**) polydispersity index, and (**C**) zeta potential of the diluted formulations. Data were expressed as mean ± SE. ***** denotes significant difference (*p* < 0.05) between the formulations.

**Figure 3 molecules-28-01354-f003:**
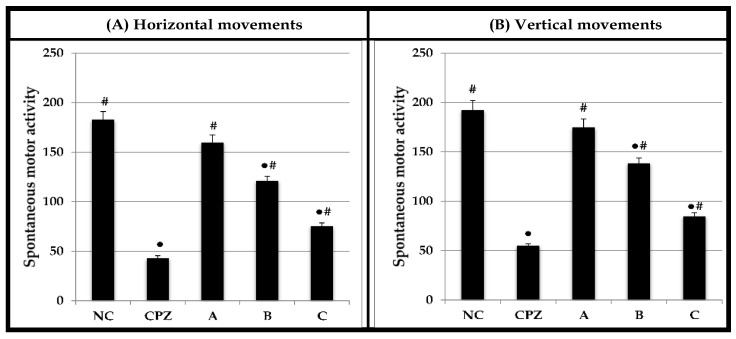
Effect of CPZ and A, B, and C formulae on spontaneous motor activity of rats for 10 min using activity cage. Values are expressed as mean ± SEM, *n* = 6 rats/ group. ● Significant change at *p* ≤ 0.05 with respect to negative control (NC) rats. # Significant change at *p* ≤ 0.05 with respect to CPZ-treated rats.

**Figure 4 molecules-28-01354-f004:**
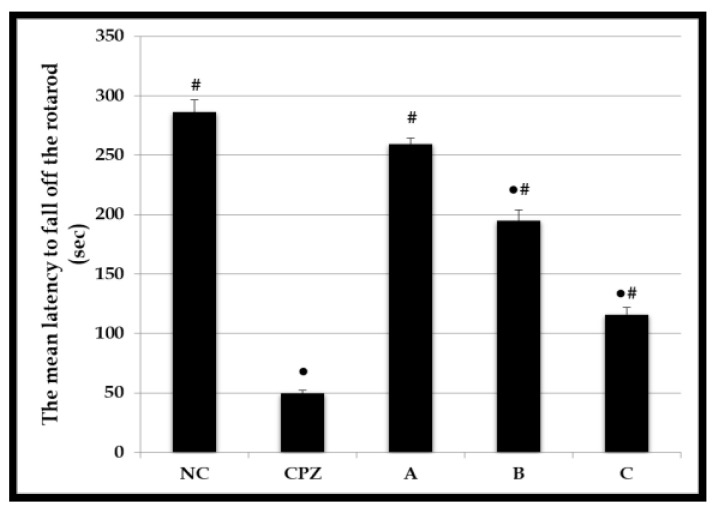
Effect of CPZ and A, B, and C formulae on motor coordination and balance in rats using accelerating rotarod. Values are expressed as mean ± SEM, *n* = 6 rats/ group. ● Significant change at *p* ≤ 0.05 with respect to negative control (NC) rats. # Significant change at *p* ≤ 0.05 with respect to CPZ-treated rats.

**Figure 5 molecules-28-01354-f005:**
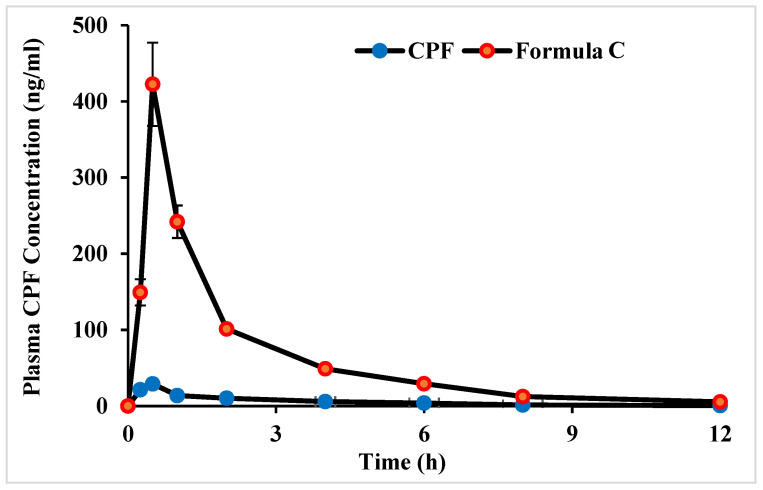
Time profiles of plasma CPF concentration (1 mg/kg) after intranasal administration of CPF solution and formula C to rats. Each data value represents the mean ± SEM of *n* = 3.

**Figure 6 molecules-28-01354-f006:**
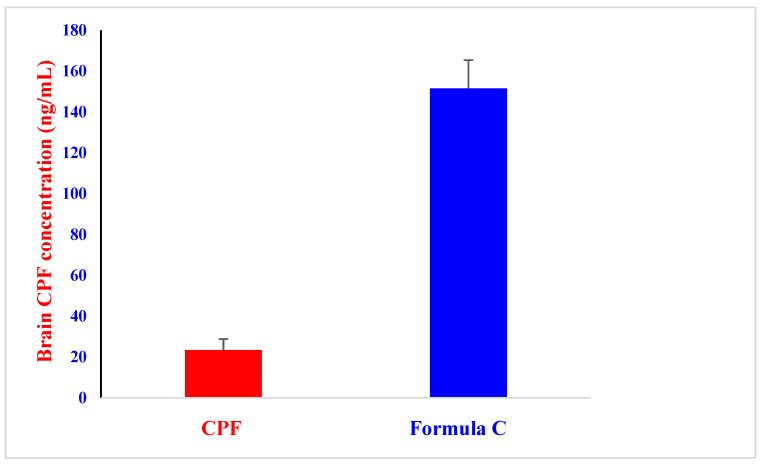
Distribution of CPF (1 mg/kg) in the brain at 15 min after intranasal administration of CPF solution and formula C to rats. Each data value represents the mean ± SEM of *n* = 3.

**Table 1 molecules-28-01354-t001:** Estimated apparent solubility of CPF in various excipients.

Excipient	Role	Estimated Apparent Solubility
Miglyol 810	Oil (medium chain triglycerides)	<2% *w*/*w*
Capmul MCM	Oil (medium chain monoglycerides	<2% *w*/*w*
Soybean oil	Oil (long chain triglycerides)	<2% *w*/*w*
Maisine 35-1	Oil (long chain monoglycerides)	<2% *w*/*w*
Caprylic acid c8	Oil (medium chain free fatty acid)	<2% *w*/*w*
Oleic acid	Oil (long chain free fatty acid)	<2% *w*/*w*
PG	Co-solvent	<2% *w*/*w*
PEG 400	Co-solvent	<2% *w*/*w*
Transcutol HP	Co-solvent	<2% *w*/*w*
DMSO	Co-solvent	20–24% *w*/*w*
Imwitor 988	Co-surfactant	<2% *w*/*w*
Imwitor 308	Co-surfactant	<2% *w*/*w*
Cremophor El	Surfactant	<2% *w*/*w*
Cremophor RH40	Surfactant	<2% *w*/*w*
Tween 85	Surfactant	<2% *w*/*w*
HCO-30	Surfactant	<2% *w*/*w*

**Table 2 molecules-28-01354-t002:** The composition (%*w*/*w*) of formulations.

Excipients	Formulations		
	A	B	C
CPF	0.5	0.5	0.5
Pluronic F-127	-	9.5	9.5
DMSO	99.5	90	30
Kolliphor El	-	-	10
Distilled water ad to	-	-	100

**Table 3 molecules-28-01354-t003:** Pharmacokinetic parameters following intranasal administration of CPF and formula C of CPF.

Pharmacokinetic Parameters	CPF Solution	Formula C
**C_max_ (ng/mL)**	29.16 ± 3.56	422.5 ± 54.71 **
**T_max_ (h)**	0.5 ± 0.0	0.5 ± 0.0
**AUC_0–12_ (ng.h/mL)**	263.02 ± 62.29	29.274.87 ± 4.855.87 **
**AUC_0-∞_ (ng.h/mL)**	265.39 ± 61.72	29.297.89 ± 4.848.26 **
**t_1/2_ (h)**	2.63 ± 0.35	2.7 ± 0.41
**K (h^−1^)**	0.27 ± 0.4	0.26 ± 0.3

Each data value represents the mean ± SEM of *n* = 3; ** *p* < 0.01, significantly different from control (CPF solution).

## Data Availability

Not applicable.

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
