# Peer review of "Brain Targeting by Intranasal Drug Delivery: Effect of Different Formulations of the Biflavone “Cupressuflavone” from Juniperus sabina L. on the Motor Activity of Rats"

_molecules, 2023, doi:10.3390/molecules28031354_

Round 1
Reviewer 1 Report
1. Formulation is not extensively characterized in terms of stability, size, PDI, loading of drug etc. 2. Optimization is not elaborated 3. Manuscript has language/grammar-related issues and needs further improvement. 4. Figures/legends are not aligned properly 5. Line no. 71, the statement seems contradictory where the author needs to clarify the meaning of powerful medication 6. Line no.129 mentioned that nanoemulsion is formed, but what was the size observed? 7. In formulation B: how the solid dispersion formation was confirmed, further why it was dissolved in DMSO.Author Response
The authors would like to thank the reviewers for the valuable comments that increase the quality of the manuscript.
Due to the addition of new figures, tables and/or references, the reviewers are kindly requested to track the movement of the figures/sections/references. Below is a summary where the comments have been addressed point-by-point, as requested by the Editor-in-Chief:
Reviewer 1
- Formulation is not extensively characterized in terms of stability, size, PDI, loading of drug etc.
According to the reviewer`s comment, a new section for formulation characterization was added to the methodology and results/discussion main sections. This section included particle size, polydispersity index and zeta potential studies.
- Optimization is not elaborated
According to the reviewer`s comment a section for "Excipient selections and formulation optimization" was added to the results and discussion section. In addition, a section for "Estimation of apparent drug solubility" was added to the methodology section.
- Manuscript has language/grammar-related issues and needs further improvement.
Intensive linguistic revision was conducted.
- Figures/legends are not aligned properly
Corrected as directed.
- Line no. 71, the statement seems contradictory where the author needs to clarify the meaning of powerful medication
Corrected as directed.
- Line no.129 mentioned that nanoemulsion is formed, but what was the size observed?
To answer this inquiry, the study of particle size analysis was added in the formulation characterization section.
- In formulation B: how the solid dispersion formation was confirmed, further why it was dissolved in DMSO.
Pluronic® F127 has surfactant-like properties and was reported to enhance the mucus penetration of nanoparticles for nose-to-brain delivery (Sonvico, Clementino et al. 2018). Accordingly, the ultimate goal of solid dispersion preparation was to homogenously load CPF into pluronic F-127 to make use of its potential nasal delivery benefits. The solid dispersion was subsequently mixed with DMSO and/or Cremophor El to produce Formulations B and C. Due to the liquid nature of these formulations, drug crystallinity in the solid dispersion might not be critical in terms of final formulation characteristics.
The newly added section " Excipients selection and formulation optimization " confirmed the ultimately high CPF solubility in DMSO. Accordingly, this excipient was involved in all three formulations to facilitate higher drug loading and minimize the risk of drug precipitation upon exposure to aqueous media.
Reviewer 2 Report
In this paper, several compounds were isolated and purified, and the activities of some compounds were tested. The paper is rich in data and records standard. All in all, this paper is good and recommended to be accepted.
Author Response
The authors would like to thank the reviewers for the valuable comments that increase the quality of the manuscript.
Due to the addition of new figures, tables and/or references, the reviewers are kindly requested to track the movement of the figures/sections/references. Below is a summary where the comments have been addressed point-by-point, as requested by the Editor-in-Chief:
Reviewer 2
In this paper, several compounds were isolated and purified, and the activities of some compounds were tested. The paper is rich in data and records standard. All in all, this paper is good and recommended to be accepted.
We thank the reviewer for the nice evaluation.
Reviewer 3 Report
The authors decribe a novel formulation containing active substances regarding the treatment of neurological diseases. The article provides a lot of supplementary materials regarding the chemical structure of the investigated compounds which are greatly detailed. Overall, the quality of the manuscript is good but some revisions are required. I find this research work important in today's drug and dosage form discovery process.
Howeve, I lack information about the optimization process of the investigated formulations. What were the criteria regarding the SNEEDS characteristics.? I also lack the description of these formulations regarding in at least simulated nasal fluids.
I would reconsider my decision after the authors comment and adjust the manuscript based on the following points:
Abstract
Line 21. – Regarding the polar fraction: please indicate in the introduction or in the results that why the polar fraction was chosen. I suppose it is due to the solubility of the investigated substances, however a question may arise: what is in the apolar fraction?
Line 27 – change alternation to alternate .
Line 31 – 32: their effect is singular, change were to was in line 32.
Introduction
Line 54 – Define alternative formulations. Are they only nanocarriers or macrocarriers?
What about the water solubility of the investigated substances? I guess they are not classified under BCS Classes yet due to lack of commercialization, but which criteria would they fit regarding the solubility and permability profile?
Results
Table 1. Headline: Change w/w% to %w/w
Table 1 – Compositions: They contain a large amount of DMSO as the water miscible solvent. Please address the toxicological concerns. Even better, find reports regarding the intranasal administration of DMSO and the safety. It is also well known that some solvents can change the permeability profile of various drugs intranasally, for example ethanol can increase it. Is it a possibility that the high amount of DMSO alter the results?
Table 1 – Please provide explanation regarding D.W.
Figure 2. – Please provide a better resolution image. Also explain NC in the text above.
Figure 3. – Please provide a better resolution image. Also explain NC in the text above.
Please indicate next to Figure 2 and 3, what is the sample (n) size.
Figure 4. Correct the legend to show what is the red dot with black line.
Figure 5. – Why only the 15 minute time point is depicted and analyzed? Can you provide a full brain concentration – time curve just like in case of the plasma concentration?
Methods
Methods are generally well-written.
Conclusion
Are there other results regarding different nanocarriers containing these drugs? Please indicate and compare if yes.
Are there other results regarding parenteral or oral administration? Please indicate and compare if yes.
Author Response
The authors would like to thank the reviewers for the valuable comments that increase the quality of the manuscript.
Due to the addition of new figures, tables and/or references, the reviewers are kindly requested to track the movement of the figures/sections/references. Below is a summary where the comments have been addressed point-by-point, as requested by the Editor-in-Chief:
Reviewer 3
The authors describe a novel formulation containing active substances regarding the treatment of neurological diseases. The article provides a lot of supplementary materials regarding the chemical structure of the investigated compounds which are greatly detailed. Overall, the quality of the manuscript is good but some revisions are required. I find this research work important in today's drug and dosage form discovery process.
However, I lack information about the optimization process of the investigated formulations. What were the criteria regarding the SNEEDS characteristics.?
I also lack the description of these formulations regarding in at least simulated nasal fluids.
According to the reviewer`s comment a section for "Excipient selections and formulation optimization" was added to the results and discussion section. In addition, a section for "Estimation of apparent drug solubility" was added to the methodology section. In addition, a new section for formulation characterization was added to the methodology and results/discussion main sections. This section included particle size, polydispersity index and zeta potential studies. The current study introduces a proof-of-concept study to evaluate the feasibility of CPF nasal delivery through the studied formulations. Future studies are required to investigate more on formulation characterization in simulated nasal fluids, chemical and physical stability.
I would reconsider my decision after the authors comment and adjust the manuscript based on the following points:
Abstract
Line 21. – Regarding the polar fraction: please indicate in the introduction or in the results that why the polar fraction was chosen. I suppose it is due to the solubility of the investigated substances, however a question may arise: what is in the apolar fraction?
The polar fractions of Juniperus species usually contain bioflavonoids, flavonoid glycosides and sugars while the non-polar fractions are rich in diterpenes and triterpenes. We selected the polar fraction for our study as we aimed to investigate the effect of compounds on the CNS. Such activities are correlated to bioflavonoids as reported in Reference 27.
Line 27 – change alternation to alternate.
Corrected as directed.
Line 31 – 32: their effect is singular, change were to was in line 32.
Corrected as directed.
Introduction
Line 54 – Define alternative formulations. Are they only nanocarriers or macrocarriers?
Corrected as directed.
Alternative formulations have been created to address the aforementioned issues, with lipidic nanosystems garnering greater attention recently as nanoemulsions, solid lipid nanoparticles, nano- lipid carriers, and self-emulsifying drug delivery systems (SEDDS) [8].
What about the water solubility of the investigated substances? I guess they are not classified under BCS Classes yet due to lack of commercialization, but which criteria would they fit regarding the solubility and permability profile?
Thanks for the reviewer’s comment.
Regarding the solubility of the drug, which is not commercialized yet, our preliminary study shows that drug is soluble in DMSO and insoluble in most of pharmaceutical solvents. Since our aim of this study is to increase the bioavailability of the drug in blood and brain bypass the BBB using SNEEDS as drug delivery system administered intranasally. Concerning the drug permeability, we proposed a new study to determine the in vitro study of drug permeability through the mucosal and epithelia membranes. Therefore, we think that drug may be BSC II or IV class which need more further examination in our future study plan.
Results
Table 1. Headline: Change w/w% to %w/w
Corrected as directed.
Table 1 – Compositions: They contain a large amount of DMSO as the water miscible solvent. Please address the toxicological concerns. Even better, find reports regarding the intranasal administration of DMSO and the safety. It is also well known that some solvents can change the permeability profile of various drugs intranasally, for example ethanol can increase it. Is it a possibility that the high amount of DMSO alter the results?
Regarding the safety of IN administration of DMSO, the following references show that DMSO have unremarkable effect on the nasal mucosa, as DMSO used as negative control in proceeding the experiments on the nasal mucosa cells of human and rats.
- Hölzer, J., Voss, B., Karroum, S., Hildmann, H. and Wilhelm, M., 2008. A comparative study of chemically induced DNA damage in isolated nasal mucosa cells of humans and rats assessed by the alkaline comet assay.Journal of Toxicology and Environmental Health, Part A, 71(13-14), pp.936-946.
- Kleinsasser, N.H., Juchhoff, J., Wallner, B.C., Bergner, A., Harréus, U.A., Gamarra, F., Bührlen, M., Huber, R.M. and Rettenmeier, A.W., 2004. The use of mini-organ cultures of human upper aerodigestive tract epithelia in ecogenotoxicology.Mutation Research/Genetic Toxicology and Environmental Mutagenesis, 561(1-2), pp.63-73.
Regarding the enhancement effect of DMSO of IN administration of drugs, DMSO is a polar aprotic solvent often used to solubilize compounds with low water solubility. Our preliminary study shows that Cupressuflavone (CPF) is insoluble in most of pharmaceutical solvents and soluble in DMSO, therefore, we use DMSO in our study as solvent for CPF. Moreover, DMSO has potential enhancer for transdermal delivery (Karakatsani, et al., Drug Dev. Ind. Pharm. 2010; Otterbach, and Lamprecht, Pharmaceutics, 2021; Salau, et al., Int. J. Pharm., 2022.), therefore, in our future study plans we need further studies to determine the enhancement permeability of DMSO on CPF through nasal epithelial membrane.
Table 1 – Please provide explanation regarding D.W.
Corrected as to Distilled water.
Figure 2. – Please provide a better resolution image. Also explain NC in the text above.
Corrected as directed.
Figure 3. – Please provide a better resolution image. Also explain NC in the text above.
Corrected as directed.
Please indicate next to Figure 2 and 3, what is the sample (n) size.
Corrected as directed.
Figure 4. Correct the legend to show what is the red dot with black line.
Corrected as directed.
Figure 5. – Why only the 15 minute time point is depicted and analyzed? Can you provide a full brain concentration – time curve just like in case of the plasma concentration?
Our aim of study is to determine the Cmax of the CPF bypass the BBB and reach the brain after intranasal administration. Therefore, we determine the maximum concertation point of CPF in brain not the full profile of brain concertation-time of the drug (Fig 5). Regarding the point time of the CPF analysis in brain, our preliminary study shows that 15 min is the maximum CPF concentration in brain after nasal administration of drug.
Methods
Methods are generally well-written.
Conclusion
Are there other results regarding different nanocarriers containing these drugs? Please indicate and compare if yes.
Until the date of the article submission there are no articles were published of nanocarriers containing the Cupressuflavone as drug delivery systems. We think that we have a potential novelty of our manuscript for functionalize the SNNEDS for IN delivery of Cupressuflavone.
Are there other results regarding parenteral or oral administration? Please indicate and compare if yes.
We are going to complete a new manuscript in effect of Cupressuflavone nanocarrier for oral administration on the motor activity in rats which we will submit it as soon as possible after revision the paper draft.
Reviewer 4 Report
1. The study emphasized the brain targeting by intranasal drug delivery, however, the study did not show the brain targeting study by IN.
2. The study investigated the different formulations to brian bypass BBB by nose-to-brain pathway, why to give the pharmacokinetic results in blood not in brain, such as hippocampus.
Author Response
The authors would like to thank the reviewers for the valuable comments that increase the quality of the manuscript.
Due to the addition of new figures, tables and/or references, the reviewers are kindly requested to track the movement of the figures/sections/references. Below is a summary where the comments have been addressed point-by-point, as requested by the Editor-in-Chief:
Reviewer 4
- The study emphasized the brain targeting by intranasal drug delivery, however, the study did not show the brain targeting study by IN.
Figure 5 shows the plasma CPF concentration at 15 min after IN administration, which indicate that IN delivery of CPF SNEEDS formula improve the distribution of CPF into the brain by 6.5 times more than drug itself. The optimized SNEEDS formula enhanced the brain concentration of CPF, which may indicate that direct transport of CPF from the nasal cavity to the brain. The findings thus revealed that formula administered intranasally could transfer CPF to the brain. Therefore, our future plan will study the CPF distribution in different brain region after IN administration of the SNEEDS formula in current study.
- The study investigated the different formulations to brian bypass BBB by nose-to-brain pathway, why to give the pharmacokinetic results in blood not in brain, such as hippocampus.
The pathways from Nose-to-Brain delivery include:
Direct transport of drug to the brain through neuronal pathways such as olfactory or trigeminal nerves. The other way is the indirect transport of drugs through the vasculature and lymphatic system, leading to the brain crossing BBB in which drug absorption from nose to brain may not be limited by one single mechanism, but may involve several pathways. (Mittal et al., Drug Deliv. 2014; Xu et al., Front. Bioeng. Biotechnol. 2020; Lee and Minko, Pharmaceutics, 2021).
Considering that CPF is known to cross the BBB due to significant increase of plasma drug concertation could lead to its distribution into the brain. Fig 4 shows that drug SNEEDS formula enhanced the systemic absorption of CPF after IN administration. Table 2 summarize the PK parameters prominent us to further examine the amount of CPF bypass the BBB and reached the brain. As shown in Fig 5 the brain’s CPF concentrations implying that IN delivery of CPF SNEEDS improve the distribution of CPF into the brain. The formula significantly enhanced the concentration of CPF, which may indicate that direct and/or direct transport of CPF from the nasal cavity to the brain which needs further study as we plan in near future.
Round 2
Reviewer 3 Report
The authors have addressed all my questions and concerns.
Please prepare figure 6 more aesthetically pleasing: the two titles for the coloumn diagram should be on the same level.
Reviewer 4 Report
The questions have been replied.